# Otoprotective Effects of Fucoidan Reduce Cisplatin-Induced Ototoxicity in Mouse Cochlear UB/OC-2 Cells

**DOI:** 10.3390/ijms24043561

**Published:** 2023-02-10

**Authors:** Cheng-Yu Hsieh, Jia-Ni Lin, Ting-Ya Kang, Yu-Hsuan Wen, Szu-Hui Yu, Chen-Chi Wu, Hung-Pin Wu

**Affiliations:** 1Department of Otolaryngology, Head and Neck Surgery, Taichung Tzu Chi Hospital, Buddhist Tzu Chi Medical Foundation, Taichung 427213, Taiwan; 2School of Medicine, Tzu Chi University, Hualien 970374, Taiwan; 3Department of Otolaryngology, Head and Neck Surgery, Hualien Tzu Chi Hospital, Buddhist Tzu Chi Medical Foundation, Hualien 970473, Taiwan; 4Institute of Medical Sciences, Tzu Chi University, Hualien 970473, Taiwan; 5Department of Music, Tainan University of Technology, Tainan 710302, Taiwan; 6Department of Otolaryngology, National Taiwan University Hospital, Taipei 100225, Taiwan; 7Department of Medical Genetics, National Taiwan University Hospital, Taipei 100225, Taiwan; 8Department of Medical Research, National Taiwan University Hospital Hsin-Chu Branch, Hsinchu 300195, Taiwan

**Keywords:** cisplatin, ototoxicity, fucoidan, antioxidant, apoptosis

## Abstract

Cisplatin is a widely used standard chemotherapy for various cancers. However, cisplatin treatment is associated with severe ototoxicity. Fucoidan is a complex sulfated polysaccharide mainly derived from brown seaweeds, and it shows multiple bioactivities such as antimicrobial, anti-inflammatory, anticancer, and antioxidant activities. Despite evidence of the antioxidant effects of fucoidan, research on its otoprotective effects remains limited. Therefore, the present study investigated the otoprotective effects of fucoidan in vitro using the mouse cochlear cell line UB/OC-2 to develop new strategies to attenuate cisplatin-induced ototoxicity. We quantified the cell membrane potential and analyzed regulators and cascade proteins in the apoptotic pathway. Mouse cochlear UB/OC-2 cells were pre-treated with fucoidan before cisplatin exposure. The effects on cochlear hair cell viability, mitochondrial function, and apoptosis-related proteins were determined via flow cytometry, Western blot analysis, and fluorescence staining. Fucoidan treatment reduced cisplatin-induced intracellular reactive oxygen species production, stabilized mitochondrial membrane potential, inhibited mitochondrial dysfunction, and successfully protected hair cells from apoptosis. Furthermore, fucoidan exerted antioxidant effects against oxidative stress by regulating the Nrf2 pathway. Therefore, we suggest that fucoidan may represent a potential therapeutic agent for developing a new otoprotective strategy.

## 1. Introduction

Hearing loss is a widely recognized common sensory deficit that affects a growing proportion of the world’s population. The World Health Organization (WHO) estimates that, by 2050, 2.5 billion people will have some degree of hearing loss, with at least 700 million requiring rehabilitation [1]. Hearing loss can be broadly classified into three types. Sensorineural hearing loss (SNHL) is the most common type of hearing loss caused by damage to the inner ear or auditory nerve, and it is becoming a serious social and health problem. Ototoxicity involves damage caused to the inner ear structures and hearing function following exposure to specific in-hospital medications, and drug-induced ototoxicity is a major side effect of several pharmacological treatments for life-threatening diseases. Among the 150 ototoxicity-associated drugs, cisplatin represents one of the most notorious and frequently used drugs in clinical practice [2]. 

Cisplatin, a highly active anticancer agent, is widely used as standard chemotherapy for various cancers [3]. However, cisplatin treatment causes severe adverse effects such as ototoxicity. SNHL caused by cisplatin treatment causes outer hair cell loss owing to apoptosis and necrosis [4], with a high incidence of 19–77% [5]. Hearing loss manifests as bilateral, serious, and generally irreversible SNHL [6,7,8]. Cisplatin-induced ototoxicity is associated with reactive oxygen species (ROS) accumulation, mitochondrial dysfunction, and apoptotic caspase activation in auditory organs [9]. Cisplatin enters hair cells through mechanoelectrical transduction channels, transporters of organic cation transporter-2, copper-like transporter-1 [10,11], as well as passive diffusion, resulting in intracellular ROS accumulation and subsequent cytotoxicity [12,13,14,15,16]. Cisplatin-induced ototoxicity is closely related to ROS generation via interactions between drugs and cochlear tissues [17,18]. Excessive toxin accumulation may lead to ROS overproduction, which triggers cochlear hair cell injury and the mitochondrial apoptotic pathway [19]. 

Cisplatin is an organometallic compound containing two amine ligands and two chloride ligands. The covalent bonds between the cisplatin platinum atom and DNA purine bases can facilitate the formation of intrastrand and interstrand crosslinks in DNA [20]. Mitochondrial DNA represents a vulnerable target of platinum. Platinated mitochondrial DNA induces excessive ROS formation and activates apoptotic cascades, eventually triggering cell death [21]. The mechanisms underlying cisplatin-induced oxidative stress in the cochlea include glutathione depletion [22], decreased antioxidant enzyme activity [23], and increased lipid peroxidation [23]. From a pathophysiological perspective, cisplatin-induced ototoxicity results in preferential high-frequency hearing impairment and outer hair cell loss. At present, there are no widely approved clinical strategies available for preventing cisplatin-induced ototoxicity. Therefore, the development of therapies to prevent ototoxicity is critical. Two mainstream protective strategies to reduce cisplatin-induced ototoxicity involve enhancing the antioxidant protection system and inhibiting the apoptotic pathway. Several antioxidants and anti-apoptotic agents can protect cells against ROS generation, such as ginkgolide [24], curcumin [25], N-acetylcysteine [26], vitamin E [25], dexamethasone [27], butyl hydroxyanisole [28], amifostine [29], allicin [30], and tertiary butyl hydroquinone [31]; however, some of these agents may be carcinogenic [32], toxic, and can inhibit the anticancer activity of cisplatin. Therefore, non-toxic, cost-effective, natural, and antioxidant compounds are in high demand.

To prevent the harmful effects of oxidative stress, it is necessary to explore novel non-toxic and exogenous antioxidants from natural sources. Previous studies have confirmed that natural plant products can reduce the effects of oxidative stress [33]. Brown seaweed extracts represent a source of natural antioxidants [34]. As a naturally active polysaccharide, fucoidan is widely studied in dietary supplements and pharmaceutical applications [35] owing to its free radical-scavenging activities [36,37]. Studies on fucoidans extracted from *Sargassum crassifolium* [38], *Sargassum cristaefolium* [39], *Sargassum hystrix* [40], *Sargassum muticum* [41], *Laminaria japonica* [36,42], and *Undaria pinnatifida* [42] have shown that fucoidan can reduce ROS accumulation. Fucoidan is a complex sulfated polysaccharide derived from the cell walls of brown seaweed and certain marine invertebrate tissues [43,44]. The structure of fucoidans varies between species. Fucoidan mainly comprises L-fucose and sulfate residues, along with uronic acid, galactose, xylose, and mannose [36]. Fucoidan is non-toxic, water-soluble, and has many biological activities that are beneficial for therapeutic applications [45]. The biological activity of fucoidan is highly dependent on its molecular weight, concentration, and sulfate content [45,46]. High-molecular-weight (>2000 kDa) fucoidans exhibit strong branching of molecules, which results in increased viscosity [47], and these molecules do not readily pass through the lipid bilayer. Low-molecular-weight (3–8 kDa) fucoidan is used more often; it can be easily absorbed through the lipid bilayer and exhibits relatively higher biological activity [48]. Fucoidan exhibits antimicrobial [49], anti-inflammatory [50], anticancer [51], anticoagulant [52], and antioxidant activities [36]. Fucoidan has been reported to effectively inhibit the growth of various cell lines in vitro [51,53,54]. In this study, we investigated the bioactivities of fucoidan derived from *Fucus vesiculosus* as a starting point for future drug development. *F. vesiculosus* is a potent source of beneficial compounds, including fucoidans, polyphenols, fucoxanthin, and essential minerals [55,56]. Fucoidans, a type of polysaccharide found exclusively in brown algae, are particularly notable [56,57]. Polysaccharides from *F. vesiculosus*, which have a primary monomer unit of α-1,3 or α-1,4 L-fucopyranosyls, mainly consist of fucose, sulfate, and ash [58,59,60,61].

Despite evidence of the antioxidant and anti-inflammatory effects of fucoidan [62], research on the otoprotective effects of fucoidan remains limited. Therefore, this study investigated the effects of fucoidan on cisplatin-induced hair cell damage and its potential otoprotective activity. Our findings suggest that fucoidan may represent a potential otoprotective agent for developing strategies to reduce cisplatin-induced ototoxicity.

## 2. Results

### 2.1. Effect of Cisplatin and Fucoidan on Cell Viability

To investigate the cytotoxicity of cisplatin in mouse cochlear UB/OC-2 cells, we initially evaluated cell viability using the 3-(4,5-dimethyl-thiazol-2-yl)-2,5-diphenyltetrazolium bromide (MTT) assay. Cisplatin significantly suppressed cell viability in a dose-dependent manner (Figure 1a). UB/OC-2 cell viability decreased to approximately 50% after treatment with 5 μM cisplatin; therefore, we used 5 μM cisplatin in subsequent experiments. In addition, we examined the cytotoxicity of fucoidan in mouse cochlear UB/OC-2 cells. Treatment with up to 2 mg/mL fucoidan alone had little or no effect on cell viability (Figure 1b). Based on these results, 2 mg/mL fucoidan was selected for subsequent experiments.

### 2.2. Fucoidan Treatment Ameliorated Cisplatin-Induced Cytotoxicity

To evaluate the effects of fucoidan on cisplatin-treated UB/OC-2 cells, the cells were treated with fucoidan, and cytotoxicity was measured based on cell viability. Cytotoxicity increased significantly in the cisplatin group and decreased in the fucoidan-treated groups compared with that in the control group (Figure 1c). Therefore, fucoidan may protect against cisplatin-induced cell damage in mouse cochlear UB/OC-2 cells.

### 2.3. Fucoidan Treatment Reduced Cisplatin-Induced Intracellular ROS Production and Reversed Mitochondrial Membrane Potential Loss

Cisplatin-induced ROS production is associated with mitochondrial dysfunction, which can induce apoptosis via the loss of mitochondrial membrane potential [22]. Intracellular ROS production, which was detected via staining with the fluorescent 2′,7′-dichlorodihydrofluorescein diacetate (DCFDA) dye, increased significantly in the cisplatin group and decreased markedly in the fucoidan-treated group (Figure 2a). To further evaluate the effect of fucoidan on cisplatin-induced oxidative damage, the mitochondrial membrane potential was analyzed using the JC-1 green/red fluorescence intensity ratio. The ratio was increased 4-fold in the cisplatin group and was significantly decreased in the fucoidan group compared with that in the control group (Figure 2b). The mitochondrial apoptotic pathway mediates the permeabilization of the outer mitochondrial membrane, resulting in mitochondrial membrane potential loss, ROS generation, and cytochrome c release from mitochondria into the cytoplasm. Cisplatin exposure induced the cytosolic accumulation of cytochrome *c*; however, this effect was attenuated by fucoidan treatment (Figure 2c). Based on these results, we suggest that fucoidan reduces cisplatin-induced intracellular ROS production and inhibits mitochondrial dysfunction.

### 2.4. Fucoidan Protects against Cisplatin-Induced Apoptosis in Mouse Cochlear UB/OC-2 Cells

To elucidate the role of fucoidan in the mitochondrial apoptotic pathway, the expression of the pro-apoptotic protein Bax and the anti-apoptotic protein Bcl-2 was analyzed. Fucoidan treatment increased the expression of Bcl-2 and decreased Bax expression at both mRNA (Figure 3a) and protein levels (Figure 3b). To determine the intrinsic apoptotic pathway involved in cisplatin-induced cell death, Western blot analysis was used to evaluate the reactive expression of apoptosis-related proteins. The expression of cleaved Caspase-9, cleaved Caspase-3, and cleaved poly (ADP-ribose) polymerase (PARP) was significantly increased in the cisplatin group but decreased in the fucoidan group (Figure 3c). These results suggested that fucoidan attenuates cisplatin-induced apoptosis in mouse cochlear UB/OC-2 cells. 

Next, we investigated the effects of fucoidan on cisplatin-mediated apoptosis via flow cytometry using Annexin V/PI double staining and the chromatin-specific dye Hoechst 33258 staining to determine intracellular cell damage. The proportion of apoptotic cells was significantly increased in the cisplatin group compared with that in the control group, whereas the proportion of apoptotic cells in the fucoidan group was lower than that in the cisplatin group (Figure 4a). Hoechst 33258 staining was performed to evaluate cell morphology and nuclear condensation in apoptotic cells. Bright blue fluorescence in apoptotic cells was markedly increased compared with that in the control group (Figure 4b). We evaluated the effect of fucoidan on the attenuation of cisplatin-induced oxidative stress and found that fucoidan significantly decreased oxidative DNA damage compared to cisplatin, based on the findings of the 8-hydroxydeoxyguanosine (8-OHdG) assay (Figure 4c). These results suggest that fucoidan can decrease cisplatin-induced apoptotic cell death. 

### 2.5. Fucoidan Attenuated Oxidative Damage and Regulated Cisplatin-Induced Apoptosis via the Nrf2 Pathway 

Fucoidan exerts an antioxidant effect via free radical-scavenging activity in a dose-dependent manner (Figure 5a). Under normal physiological conditions, nuclear factor erythroid 2-related factor 2 (Nrf2) binds to Kelch-like ECH-associated protein 1 (keap1) and destabilizes in the cytoplasm; however, under oxidative stress conditions, the binding between Keap1 and DLG is disrupted, and Nrf2 rapidly transfers and accumulates into the nucleus, thereby activating the transcription of downstream antioxidant enzyme genes. Fucoidan treatment significantly increased the protein expression level of Nrf2 in the nuclear fraction (Figure 5b). UB/OC-2 cells were treated with various concentrations of fucoidan (0.25–2 μg/mL) under cisplatin-induced oxidative stress conditions, and they exhibited significantly increased *Nrf2* mRNA levels and downstream mRNA expression products of heme oxygenase-1 (*Ho1*) and NAD(P)H Quinone Dehydrogenase 1 (*Nqo1*) (Figure 5c). Protein expression analysis of fucoidan-treated UB/OC-2 cells demonstrated significant increases in the protein expression levels of Nrf2 and antioxidant enzymes (HO-1 and NQO1) under oxidative stress conditions (Figure 5d). Moreover, fucoidan treatment significantly increased the protein expression levels of other antioxidant enzymes, including superoxide dismutase 1 (SOD1), SOD2, catalase, and glutathione peroxidase (GPx), compared with those in the cisplatin group (Figure 5e). Taken together, our results suggest that fucoidan may exert antioxidant effects against oxidative stress via Nrf2-regulated antioxidant enzymes.

## 3. Discussion

To our knowledge, this is the first study to explore the otoprotective effects of fucoidan against cisplatin-induced ototoxicity. Previous studies have demonstrated the strong ROS-scavenging activity of fucoidan; thus, the use of non-toxic natural compounds may represent a feasible strategy for the prevention or treatment of diseases caused by oxidative stress. In the present study, cisplatin treatment decreased cell viability, enhanced cytotoxicity, stimulated ROS generation, reduced mitochondrial membrane potential, and eventually led to apoptotic cell death in mouse cochlear UB/OC-2 cells. Fucoidan treatment protected mouse cochlear UB/OC-2 cells against cisplatin-induced cell death by ameliorating ROS production and suppressing the mitochondrial apoptotic signaling pathway.

At present, cisplatin-induced ototoxicity is believed to be mainly caused by excess ROS formation. To investigate the antioxidant effect of fucoidan on cisplatin-induced oxidative stress, DPPH, 8-OHdG, DCFDA, and JC-1 staining were performed to evaluate the antioxidant properties. Fucoidan increased the viability of mouse cochlear UB/OC-2 cells in a dose-dependent manner (Figure 1c). Based on cell viability, intracellular ROS levels, and apoptotic body formation in mouse cochlear UB/OC-2 cells, our results showed that fucoidan greatly reduces cisplatin-induced ROS production (Figure 4b). Moreover, we found that fucoidan attenuates oxidative damage and regulates cisplatin-induced apoptosis by regulating the Nrf2 pathway. Taken together, our findings suggest that fucoidan may represent a potential otoprotective therapeutic agent.

The factors that determine the antioxidant activity of fucoidan are comprehensive and do not represent a single factor [63]. The factors include concentration, molecular weight, thiol composition, glycosidation branching, algal species, and the extraction method used [37,63,64,65]. Regarding size, low-molecular-weight fucoidan displays relatively higher biological activity owing to its lipid bilayer [66]. Yuan et al. showed that the reduced ability of fucoidan increases with an increase in molecular weight [65]. Philipp et al. indicated that high-molecular-weight (>2000 kDa) fucoidans are more effective in protection against oxidative stress and may interact with cellular pathways [67,68]. In our study, the tested extracts represented high-molecular-weight fucoidans that reduced cisplatin-induced intracellular ROS production in mouse cochlear UB/OC-2 cells. In addition to molecular weight, the substituents of fucoidan play a critical role in antioxidant activity [69]. Overall, a positive correlation has been reported between the levels of sulfate groups and antioxidant activity [47]. Moreover, the sulfate content ratio may greatly contribute to radical-scavenging ability [44]. In addition to sulfate content and molecular weight, the presence of polyphenols also shows a significantly high correlation with stronger free radical-scavenging activity [42,70,71].

Antioxidants and anti-apoptotic agents are thought to attenuate the harmful effects of ROS and may be effective in treating oxidative stress-related diseases. In addition to antioxidant capacity, fucoidan exhibited anti-apoptotic potential via the mitochondrion-mediated caspase activation pathway in our study. ROS accumulation may impair RNA translation, protein synthesis, and mitochondrial membrane permeability, resulting in mitochondrion-dependent apoptosis. To further evaluate the anti-apoptotic activity of fucoidan, we tested the expression of apoptosis-related proteins by performing Annexin V/PI double staining and Hoechst 33258 staining. The balance between the pro-apoptotic protein Bax and the anti-apoptotic Bcl-2 protein is crucial for cell survival. Oxidative stress creates an imbalance that favors apoptosis. Bax first translocates from the cytoplasm to the mitochondria and triggers mitochondrial outer membrane permeabilization, resulting in the loss of the mitochondrial membrane potential and the release of cytochrome c from mitochondria into the cytoplasm [72]. In our study, cisplatin treatment resulted in the cytosolic accumulation of cytochrome *c*; however, treatment with fucoidan arrested cytochrome *c* inside the mitochondrial membrane, inhibiting the apoptosis pathway. Our results demonstrate that fucoidan may inhibit apoptosis by stabilizing the mitochondrial membrane potential and inhibiting mitochondrial apoptotic cascades, thereby protecting cochlear cells against cisplatin-induced cell death. Overall, we found that fucoidan possesses both antioxidant and anti-apoptotic activities. 

ROS can stimulate cochlear inflammation, suggesting that the use of anti-inflammatory agents may be required for the treatment of hearing loss [41]. Cisplatin is clinically administered to patients with cancer who generally exhibit infection or inflammation; therefore, the impact of treatment with otoprotective drugs and the severity of ototoxicity during the inflammatory response should be fully investigated. Inflammation can change cochlear physiology and induce ROS production by activating extracellular signal kinase and nuclear factor kappa-light-chain-enhancer of activated B cells. NADPH oxidase 3 plays a crucial role in ROS generation in the cochlear region, triggering an inflammatory process via the activation of signal transducer and transcription factor activator 1 (STAT1) [73]. STAT is an important mediator of cell death that stimulates apoptosis. Several studies have demonstrated the critical role of anti-inflammatory ROS scavengers. Kaur et al. indicated that short-interfering RNA against STAT1 attenuates cisplatin-induced ototoxicity by suppressing inflammation [74]. Fucoidan can significantly inhibit the release of nitric oxide induced by bacterial lipopolysaccharide and reduce inflammation [37]. Fucoidan treatment for meningitis in rats reduces all inflammatory changes [75]. Nrf2 also plays a central role in maintaining intracellular redox homeostasis and subsequently regulates anti-inflammatory functions [76]. Our results suggest that fucoidan may have protective effects against oxidative stress via Nrf2-regulated antioxidant enzymes. Therefore, fucoidan may represent a promising otoprotective agent owing to its anti-inflammatory, antioxidant, and anti-apoptotic properties. 

Fucoidan, which has been used as a dietary supplement for years, has received regulatory approval from various global authorities for use in foods and supplements. Specifically, fucoidan extracts from *F. vesiculosus* have been deemed “Generally Recognized as Safe” by the US FDA and have been approved as novel foods by the European Commission. These extracts have also been approved for use in Canada and Australia [56]. The recommended daily consumption is up to 250 mg/day (4.17 mg/kg body weight (bw)/day for a 60 kg person) [77]. The study focuses on the use of fucoidan extracts from *F. vesiculosus* as a potential treatment for cisplatin-induced ototoxicity. In an in vivo study conducted on Sprague Dawley rats, no significant side effects were observed following oral administration of 0–1000 mg/kg of fucoidan. However, a significant elevation in plasma ALT was observed when the concentration was increased to 2000 mg/kg [78]. In the study by Naoki et al., the participants consumed five capsules containing 166 mg of fucoidan daily for a period of up to 12 months. No significant adverse reactions were reported among any of the participants [79]. In a study by Hsiang et al., the effectiveness of fucoidan as a complementary therapy for chemotherapy and targeted drugs in patients with metastatic colorectal cancer was evaluated. Patients who consumed 4 g of fucoidan daily showed a significant improvement in disease control rate [80]. 

Fucoidan is a natural product that can vary in chemical composition based on the species and extraction method, so the quality and purity should be carefully evaluated before use. However, its use in pregnant women, breastfeeding women, and those with poor liver and kidney function is contraindicated. A small number of people may experience mild diarrhea, which usually disappears a few days after consumption. Fucoidan may also have blood thinning properties; hence, its use is contraindicated in patients with poor coagulation function and those on anticoagulant therapy, and should also not be consumed before surgery. Although the iodine content of fucoidan extract is substantially lower than that of seaweed, people who need to limit their iodine intake should still be cautious. Future studies are needed to determine the safety and efficacy of fucoidan to reduce cisplatin-induced ototoxicity.

Our findings suggest that fucoidan may have a protective effect against cisplatin-induced ototoxicity, but this was only observed in the UB/OC-2 cell line. Therefore, further research is needed to confirm these results. Our future studies will aim to determine the safety and efficacy of fucoidan in reducing cisplatin-induced ototoxicity using animal models. The main focus of our study was to initially evaluate the protective effects of fucoidan in preventing ototoxicity rather than as a treatment. In this study, we used a commercial fucoidan product with 85% purity. In our future studies, the protective effect of fucoidan with different molecular weights, different fucoidan species, and levels of purity will be explored. Additionally, the most effective and safe way to administer the drug will be further studied using animal models.

## 4. Materials and Methods

### 4.1. Cell Culture

The UB/OC-2 cell line was purchased from Ximbo (London, UK) and cultured in minimal essential medium/GlutaMAX (Gibco, New York, NY, USA) supplemented with 10% fetal bovine serum (Hyclone Laboratories Inc., Logan, UT, USA) and 50 U/mL interferon-gamma (R&D Systems, Minneapolis, MN, USA). Cells were maintained in a humidified atmosphere containing 5% CO_2_ and 95% O_2_ at 33 °C. Cisplatin was purchased from Fresenius Kabi Oncology Ltd. (Taipei, Taiwan). Commercial fucoidan extracts used in this experiment derived from *F. vesiculosus* were purchased from Da Yi Biotech & Health Food Co., Ltd. (Chiayi, Taiwan), which has good manufacturing practice certification. The raw materials have received the Australian Organic Certification (ACO: 11011) and the products also have the Taiwan Compassionate Organic Certification (TOC-AP0003 GMP) and ISO 9001 certification. Each bag contains 1 g of powder, with a fucoidan purity of 85% and an approximate sodium content of 0.011 g. It also contains 0.7 g of carbohydrates, 0.05 g of iodine, and 0.02 g of protein. The main chemical components are fucose, xylose, sulfate, and ash [60]. To evaluate the protective effect of fucoidan against cisplatin-induced cytotoxicity, cells were exposed to 5 μg/mL cisplatin for 24 h after pretreatment with 2 mg/mL fucoidan for 2 h.

### 4.2. Cell Viability Assay

Cell viability was quantified using MTT (Enzo Life Sciences, Inc., Farmingdale, PA, USA) reduction assay. The cells were pre-incubated overnight in 24-well plates at a density of 6 × 10^4^ cells/well. Cisplatin and/or fucoidan were added at various concentrations after 24 h. After the cells were subjected to different treatments, MTT solution was added at a final concentration of 0.2 mg/mL for 4 h at 33 °C, and the formazan crystals were dissolved in 200 μL dimethyl sulfoxide. The absorbance was measured at 570 nm using a microplate reader (Infinite 200 PRO Series Multimode Reader; Tecan, Männedorf, Switzerland).

### 4.3. Detection of ROS Production

Intracellular ROS levels were measured using the fluorescent dye DCFDA (Enzo Life Sciences). The cells were pre-incubated overnight in 12-well plates at a 1.7 × 10^5^ cells/well density. After various treatments, the cells were incubated with DCFDA solution at a final concentration of 20 μM for 30 min at 33 °C. Stained cells were rinsed with phosphate-buffered saline (PBS), dislodged from the culture dish, and resuspended in PBS. Fluorescence intensity was measured at an excitation wavelength of 485 nm and an emission wavelength of 535 nm using a BD Accuri C6 flow cytometer (BD Biosciences, San Jose, CA, USA). 

### 4.4. Detection of Mitochondrial Membrane Potential 

Changes in mitochondrial membrane potential were monitored via flow cytometry using 5,5′,6,6′-tetrachloro1,1′,3,3′-tetramethylbenzimidazolylcarbocyanine iodide (JC-1; Enzo Life Sciences) dye. The cells were pre-incubated overnight in 6 cm dishes (1 × 10^6^ cells). After various treatments, the cells were incubated with 5 μg/mL JC-1 in a culture medium for 15 min at 33 °C. The stained cells were rinsed with PBS, dislodged from the culture dish, and resuspended in PBS. Fluorescence signals were detected at an excitation wavelength of 515 nm and an emission wavelength of 530/590 nm using a BD Accuri C6 flow cytometer (BD Biosciences). Changes in mitochondrial membrane potential were expressed as the ratio of green fluorescence intensity to red fluorescence intensity. 

### 4.5. Assessment of Cytosolic, Nuclear, and Mitochondrial Fractionation

The cytoplasmic and nuclear fractions were separated using a subcellular fractionation kit (Abcam, Cambridge, MA, USA) according to the manufacturer’s instructions. The cells were pre-incubated overnight in 10 cm dishes (2.5 × 10^6^ cells). After various treatments, the cells were harvested and resuspended in Buffer A (2 × 10^7^ cells/μL; 0.015% EDTA and 0.36% Tris). Equal volumes of Buffer B (0.015% EDTA, 0.36% Tris, and 0.001% digitonin) were added to the tube following incubation for 7 min at room temperature (25–28 °C). After centrifuging the samples at 10,000× *g* for 1 min at 4 °C, the supernatant comprising cytosolic proteins was obtained. The pellet was resuspended in the original volume of Buffer A and the same amount of Buffer C (0.03% EDTA and 0.75% Tris) was added for 10 min at room temperature. After centrifuging the samples at 10,000× *g* for 1 min at 4 °C, the supernatant comprising mitochondrial proteins was obtained. The pellet was resuspended in the original volume of Buffer A and the lysate was centrifuged at 16,000× *g* for 10 min at 4 °C; the supernatant comprising nuclear proteins was obtained. Subcellular fractions were stored at −80 °C until they were used for Western blot analysis. 

### 4.6. Western Blot Analysis

The cells were pre-incubated overnight in 6 cm dishes (1 × 10^6^ cells). After various treatments, the cells were collected and lysed using radioimmunoprecipitation assay lysis buffer (Thermo Fisher Scientific, Waltham, MA, USA) containing protease (EMD Millipore, Darmstadt, Germany) and phosphatase inhibitors (EMD Millipore). The total protein concentration was quantified using a bicinchoninic acid reagent (VWR International LLC, Radnor, PA, USA). Cell lysates (30 μg/lane) or subcellular fractions were separated on 8–15% sodium dodecyl sulfate-polyacrylamide gels and then transferred to polyvinylidene fluoride membranes (Millipore, Burlington, MA, USA). The membrane was blocked in 5% skim milk in Tris-buffered saline for 1 h at room temperature and incubated overnight at 4 °C with primary antibodies: Bcl-2 (1:1000; Cell Signaling Technology, Beverly, MA, USA), Bax (1:1000; Cell Signaling Technology), Caspase-9 (1:1000; Cell Signaling Technology), Caspase-3 (1:1000; Cell Signaling Technology), PARP (1:1000; Cell Signaling Technology), Nrf2 (1:1000, Abcam), HO-1 (1:1000; Proteintech, Manchester, UK), NQO1 (1:1000; Abcam), SOD1 (1:10,000; Abcam), SOD2 (1;1000; Elabscience Biotechnology Inc., Houston, TX, USA), catalase (1:1000; Elabscience Biotechnology Inc.), and GPx (1:1000; Elabscience Biotechnology Inc.). The membrane was then incubated with a 1:5000-diluted horseradish peroxidase-labeled secondary antibody (PerkinElmer Life Science, Boston, MA, USA) for 1 h at room temperature. The results were obtained using the enhanced chemiluminescence detection kit (Bio-Rad Laboratories, Inc., Hercules, CA, USA), and images were detected using the KETA C Chemi Image System (Wealtec Corporation, Sparks, NV, USA). Images were quantified by densitometry analysis using Image J software in version 1.54a (National Institutes of Health) and normalized to the corresponding level of β-actin in each sample. Fold change values were then calculated as the relative abundance of a target protein to the control sample on the same membrane.

### 4.7. Reverse Transcription-Quantitative Polymerase Chain Reaction 

Detection and quantification of mRNA gene expression were performed using reverse transcription-quantitative polymerase chain reaction (RT-qPCR). Total RNA was isolated using the RNeasy Mini Kit (Qiagen GmbH, Hilden, Germany) and QIAshredder (Qiagen GmbH), according to the manufacturer’s instructions. Equal amounts of RNA (1 μg) were converted to cDNA in a total volume of 20 μL using the SuperScript Reverse Transcriptase kit (Thermo Fisher Scientific). Gene expression was evaluated using an SYBR Green PCR kit (Qiagen, Hilden, Germany) and a StepOnePlus Real-Time PCR system (Thermo Fisher Scientific). The primer sequences were as follows: *Bcl2*: forward 5′-GTGGATGACTGAGTACCTGAACC-3′ and reverse 5′-AGCCAGGAGAAATCAAACAGAG-3′; *Bax*: forward 5′-CCGAGAGGTCTTTTTCC-3′ and reverse 5′-GCCTTGAGCACCAGTTTG-3′; *Nrf2*, forward 5′-CATGATGGACTTGGAGTTGC-3′ and reverse 5′-CCTCCAAAGGATGTCAATCAA-3′; for *Ho1*, forward 5′-GGTGATAGAAGAGGCCAAGA-3′ and reverse 5′-AGCTCCTGCAACTCCTCAAA-3′; for *Nqo1*, forward 5′-AGCGTTCGGTATTACGATCC-3′ and reverse 5′-AGTACAATCAGGGCTCTTCTCG-3′; and *Gapdh*, forward and reverse. The relative fold-change was calculated using the comparative 2^−ΔΔCt^ method (ΔCt = Ct [target gene] − Ct [GAPDH]).

### 4.8. Determination of Apoptotic Cell Death 

The extent of apoptotic cell death in the different treatment groups was determined using the Annexin V-FITC Apoptosis Detection Kit (Elabscience Biotechnology Inc.). The cells were pre-incubated overnight in 6 cm dishes (1 × 10^6^ cells). After various treatments, the cells were harvested and resuspended in 400 μL binding buffer containing Annexin V-FITC and propidium iodide (PI) according to the manufacturer’s instructions. The cell population in each group was measured using a BD Accuri C6 flow cytometer (BD Biosciences). 

### 4.9. Detection of DNA Condensation

Hoechst 33258 dye was used to monitor apoptotic cells that contained condensed or fragmented nuclei. The cells were pre-incubated overnight in six-well plates at a 3.8 × 10^5^ cells/well density. After various treatments, the cells were permeabilized in 100% methanol for 20 min at −20 °C and then incubated with 20 μg/mL Hoechst 33258 (Enzo Life Sciences) for 20 min at 33 °C. Fluorescence signals were detected at an excitation wavelength of 352 nm and an emission peak of 454 nm using an Olympus BX41 microscope (Tokyo, Japan). 

### 4.10. Detection of Oxidative DNA Damage 

Oxidative DNA damage was determined based on the 8-OhdG content. Cells were pre-incubated overnight in 96-well plates at a density of 9 × 10^3^ cells/well. After the cells were subjected to different treatments, the culture supernatant was collected, and oxidative DNA damage was analyzed using an 8-OhdG enzyme-linked immunosorbent assay kit (Elabscience Biotechnology Inc.) according to the manufacturer’s protocol. Absorbance was measured at an absorption wavelength of 450 nm using a microplate reader (Infinite 200 PRO Series Multimode Reader).

### 4.11. DPPH Free Radical-Scavenging Assay 

The free radical-scavenging activity of fucoidan was evaluated using a DPPH assay. Briefly, 20 μL of fucoidan at various concentrations (0.25, 0.5, 1, and 2 μg/mL) was mixed with 80 μL Tris-HCl (100 mM Tris-HCl, pH 7.4), and 100 μL methanolic solution of DPPH was added separately. After 20 min of incubation in the dark at room temperature, absorbance was measured at 517 nm using a microplate reader (Infinite 200 PRO Series Multimode Reader). The results were calculated using the following equation:DPPH free radical-scavenging activity (%) = [1 − (As/Ac)] × 100.
where Ac is the absorbance value of the control and As is the absorbance value of the sample.

### 4.12. Statistical Analysis

Statistical analysis was performed using SPSS Version 22.0 software (IBM Corporation, Armonk, NY, USA). Data are expressed as the mean ± standard deviation (SD), and statistical differences among groups were assessed using one-way analysis of variance with post-hoc Tukey’s test. Values of *p* < 0.05 were considered significant.

## 5. Conclusions

Fucoidan could protect mouse cochlear UB/OC-2 cells against cisplatin-induced cell death by ameliorating ROS production, suppressing the mitochondrial apoptotic pathway, and activating antioxidant enzymes. Owing to its antioxidant, anti-apoptotic, non-toxic, and natural properties, fucoidan represents a novel therapeutic agent with otoprotective effects. 

## Figures and Tables

**Figure 1 ijms-24-03561-f001:**
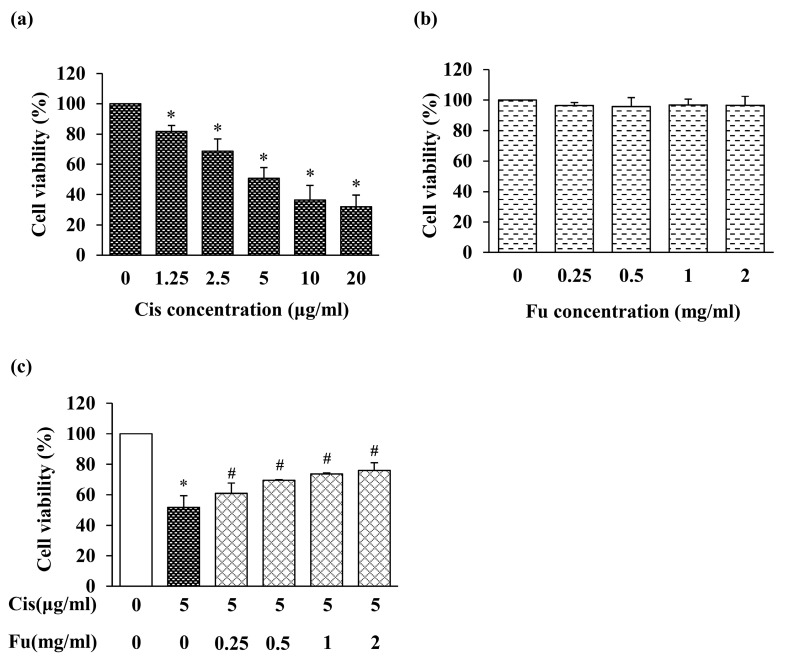
Effects of Cis and Fu on the viability of UB/OC-2 cells based on MTT assays. Cells were exposed to Cis (1.25–20 μg/mL) (**a**) and Fu (0.25–2 mg/mL) (**b**) for 24 h. (**c**) Effects of Fu on cell viability after Cis stimulation. Cells were cultured with Fu at the indicated concentration for 2 h, followed by stimulation with Cis (5 μg/mL) for 24 h. The results are expressed as mean ± SD. n = 5 per group. * *p* < 0.05 vs. the control group; # *p* < 0.05 vs. the cisplatin group. Fu, fucoidan; Cis, cisplatin.

**Figure 2 ijms-24-03561-f002:**
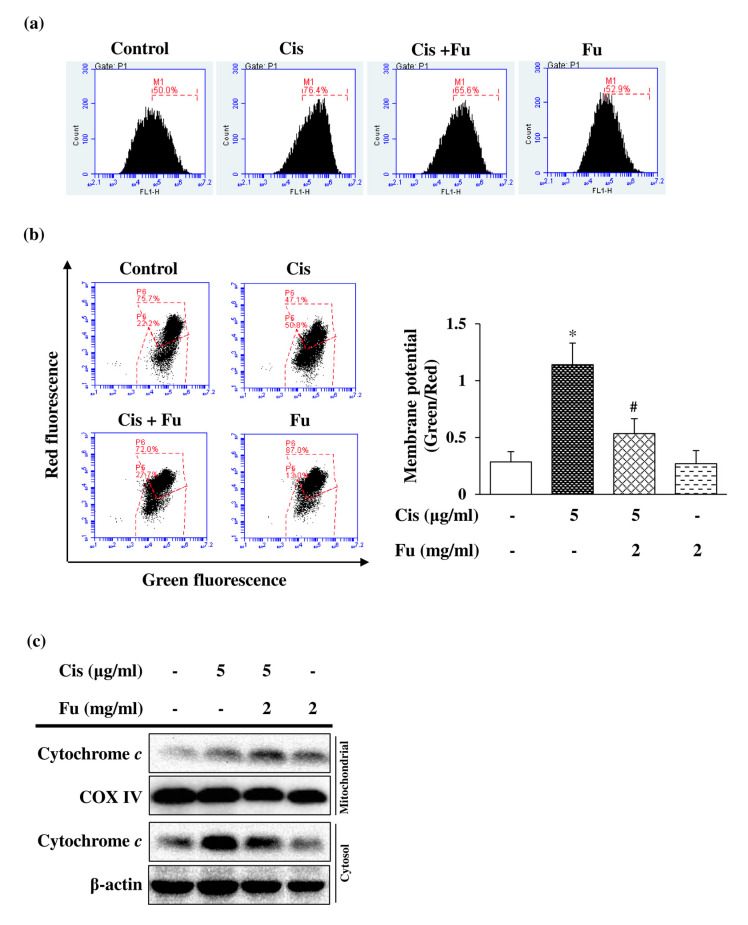
Effects of Fu on Cis-induced ROS production and mitochondrial dysfunction in UB/OC-2 cells. Cells were cultured with Fu (2 mg/mL) for 2 h, followed by stimulation with Cis (5 μg/mL) for 24 h. (**a**) ROS levels were monitored using the fluorescent dye 2′,7′-dichlorodihydrofluorescein diacetate via flow cytometry. (**b**) Mitochondrial membrane potential was detected via flow cytometry using the fluorescent dye 5,5′,6,6′-tetrachloro1,1′,3,3′-tetramethylbenzimidazolylcarbocyanine iodide. The ratio of the green fluorescence intensity to red fluorescence intensity was quantified. The results are expressed as the mean ± standard deviation. n = 5 per group. * *p* < 0.05 vs. the control group; # *p* < 0.05 vs. the Cis group. (**c**) Expression levels of cytochrome *c* in the mitochondrial and cytosolic fractions were measured using Western blot analysis. COX IV and β-actin were used as internal controls for mitochondrial and cytosolic fractions, respectively. Fu, fucoidan; Cis, cisplatin; ROS, reactive oxygen species; COX IV, cytochrome *c* oxidase (complex IV).

**Figure 3 ijms-24-03561-f003:**
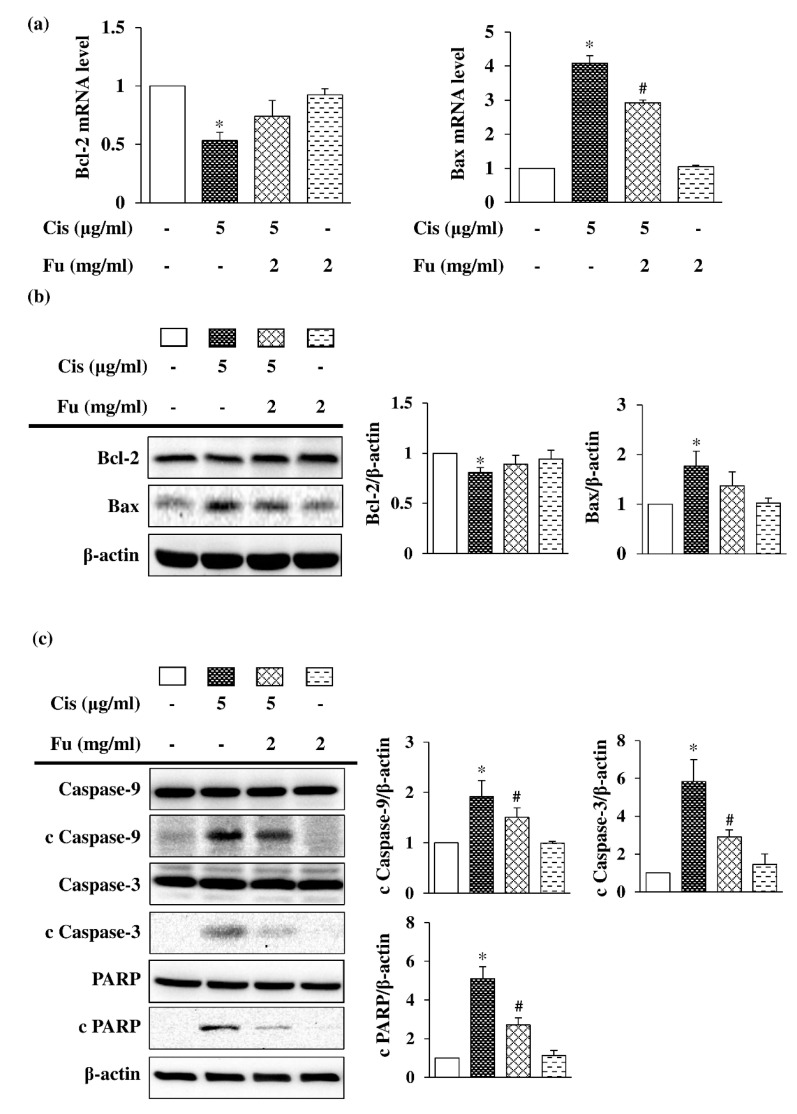
Effects of Fu on Cis-induced apoptosis in UB/OC-2 cells. Cells were cultured with Fu (2 mg/mL) for 2 h, followed by stimulation with Cis (5 μg/mL) for 24 h. (**a**) The mRNA expression levels of Bcl-2 and Bax were determined via a reverse transcription-quantitative polymerase chain reaction. (**b**) Western blot analysis was used to determine Bcl-2 and Bax expression levels. (**c**) Western blot analysis was used to determine the expression level of apoptosis-related proteins, Caspase-9, cleaved Caspase-9, Caspase-3, cleaved Caspase-3, PARP, and cleaved PARP. β-actin was used as the loading control. The results are expressed as the mean ± standard deviation. n = 3 per group. * *p* < 0.05 vs. the control group; # *p* < 0.05 vs. the cisplatin group. Fu, fucoidan; Cis, cisplatin; PARP, poly (ADP-ribose) polymerase.

**Figure 4 ijms-24-03561-f004:**
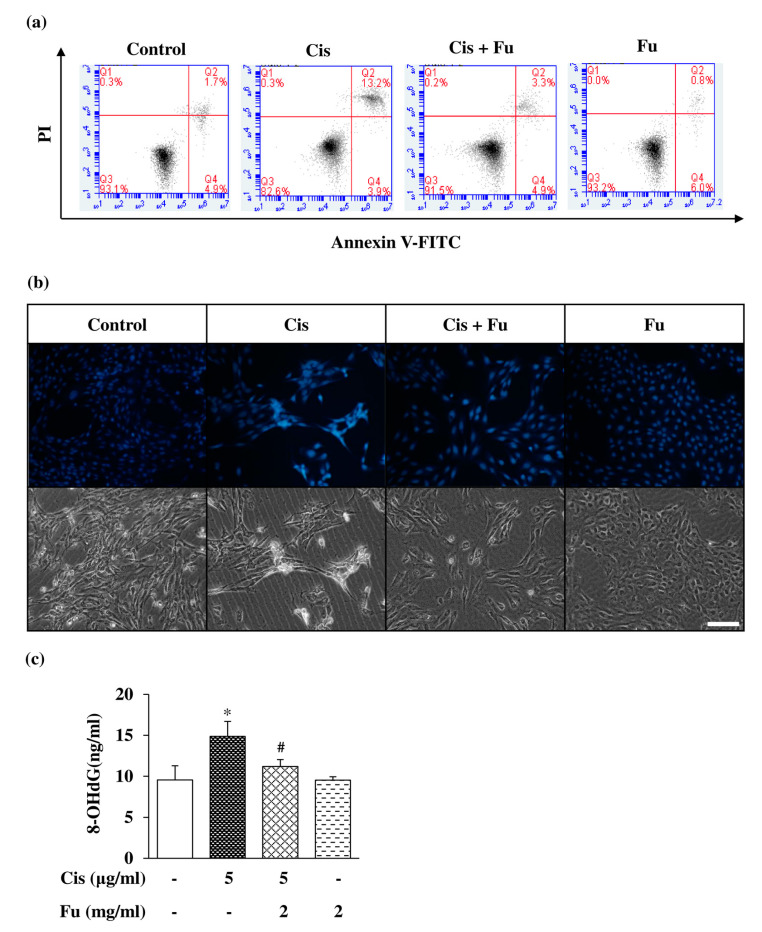
Effects of Fu on Cis-induced apoptosis of UB/OC-2 cells. Cells were cultured with Fu (2 mg/mL) for 2 h, followed by stimulation with Cis (5 μg/mL) for 24 h. (**a**) Apoptotic cells were analyzed using an Annexin V-FITC Apoptosis Detection Kit via flow cytometry. (**b**) Apoptotic nuclei that exhibited condensed or fragmented DNA were stained with Hoechst 33258 and observed under a fluorescence microscope (×200). Scale bar =100 μm. (**c**) The extent of oxidative DNA damage was assessed using an 8-OHdG ELISA Kit. The results are expressed as the mean ± SD. n = 3 per group. * *p* < 0.05 vs. the control group; # *p* < 0.05 vs. the cisplatin group. Fu, fucoidan; Cis, cisplatin; 8-OHdG, 8-hydroxy-2-deoxyguanosine; PI, propidium iodide; FITC, fluorescein isothiocyanate.

**Figure 5 ijms-24-03561-f005:**
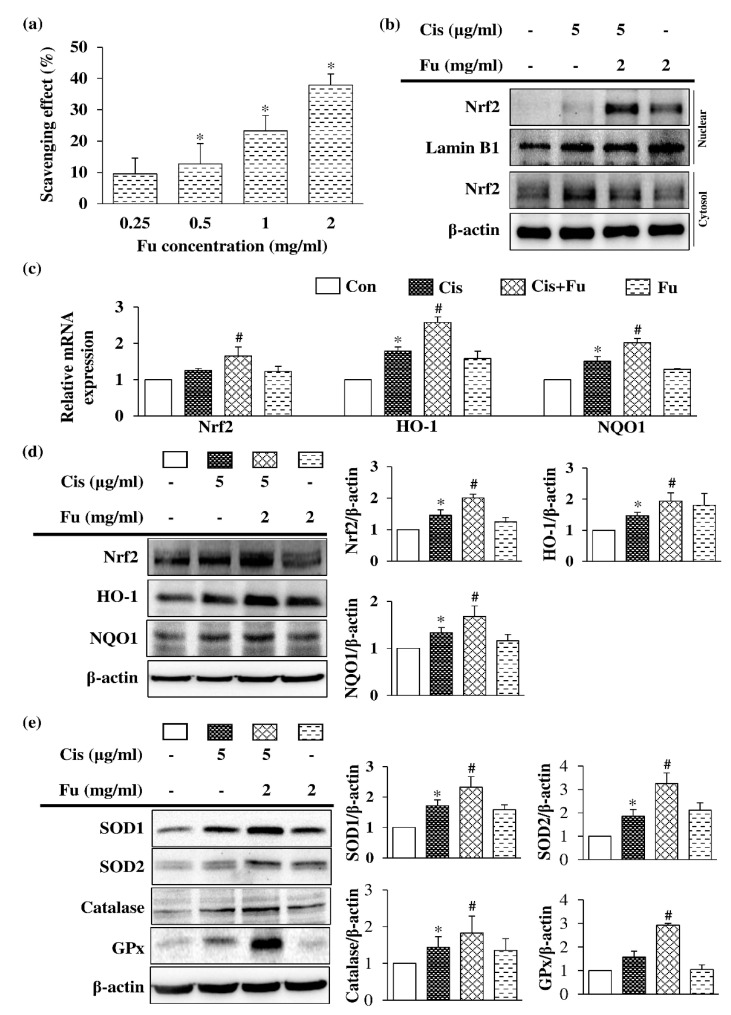
Effects of Fu on the antioxidant response in Cis-treated UB/OC-2 cells. (**a**) The free radical-scavenging activity of Fu (0.25–2 mg/mL) was determined via DPPH assays. Cells were cultured with Fu (2 mg/mL) for 2 h, followed by stimulation with Cis (5 μg/mL) for 24 h. (**b**) Protein expression levels of Nrf2 in the nuclear and cytosolic fractions were measured via Western blot analysis. Lamin B1 and β-actin were used as internal controls for nuclear and cytosolic fractions, respectively. (**c**) The mRNA expression levels of *Nrf2*, *Ho1*, and *Nqo1* were determined using reverse transcription-quantitative polymerase chain reaction. (**d**) Determination of Nrf2, HO-1, and NQO1 protein expression levels was based on Western blotting analysis. (**e**) Expression of SOD1, SOD2, catalase, and GPx protein expression levels were determined using Western blot analysis. β-actin was used as a loading control. The results are expressed as the mean ± standard deviation. n = 3 per group. * *p* < 0.05 vs. the control group; # *p* <0.05 vs. the Cis group. Fu, fucoidan; Cis, cisplatin; Nrf2, NF-E2–related factor 2; HO-1, heme oxygenase 1; NQO1, NAD(P)H Quinone Dehydrogenase 1; SOD1, superoxide dismutase 1; SOD2, superoxide dismutase 2; GPx, glutathione peroxidase.

## Data Availability

Not applicable.

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
