# Peer review of "Otoprotective Effects of Fucoidan Reduce Cisplatin-Induced Ototoxicity in Mouse Cochlear UB/OC-2 Cells"

_ijms, 2023, doi:10.3390/ijms24043561_

Round 1
Reviewer 1 Report
Overall, this manuscript describes a well conducted study. However, although the title describes the study, is too long. It is worthy to search novel natural resources as potential therapeutic agents.
Author Response
Response to reviewer 1:
|
Comments |
Responses |
|
Overall, this manuscript describes a well conducted study. However, although the title describes the study, is too long. It is worthy to search novel natural resources as potential therapeutic agents.
|
Thank you for your valuable suggestions. We have made the necessary changes in the revised manuscript as shown below
Title: Otoprotective effects of Fucoidan Reduce Cisplatin-Induced Ototoxicity in Mouse Cochlear UB/OC-2 Cells |
Reviewer 2 Report
1. This study lacks animal experiments, so it is difficult to estimate the actual effect of Fucoidan in relieving cisplatin ototoxicity.
2. In fact, Fucoidan is first protected and then treated with cisplatin, lacking evaluation of the therapeutic effect of Fucoidan.
Author Response
Response to reviewer 2:
|
Comments |
Responses |
|
1. This study lacks animal experiments, so it is difficult to estimate the actual effect of Fucoidan in relieving cisplatin ototoxicity. |
Thank you very much for the valuable comment. Indeed, it would be intriguing to investigate the effects of fucoidan in future studies. It is crucial to note that our results obtained in an in vitro model should be confirmed using animal or ex vivo models. Our future research will focus on validating these results using animal models. We agree with the reviewer that the lack of animal experiments is a limitation of this study, and we have elaborated on the limitation in the revised manuscript.
Pages 11, lines 253-254, page 12-13, lines 358-367 To our knowledge, this study is the first to investigate the potential otoprotective effects of fucoidan on mouse cochlear cells.
Our findings suggest that fucoidan may have a protective effect against cisplatin-induced ototoxicity, but this was only observed in the UB/OC-2 cell line. Therefore, further research is needed to confirm these results. Our future studies will aim to determine the safety and efficacy of fucoidan in reducing cisplatin-induced ototoxicity using animal models. The main focus of our study was to initially evaluate the protective effects of fucoidan in preventing ototoxicity rather than as a therapeutic treatment. In this study, we used a commercial fucoidan product that was 85% pure. We plan to explore the protective effect of fucoidan with different molecular weights, different fucoidan species, and levels of purity. Additionally, it is important to consider the most effective and safe way to administer the drug, which will be further studied using animal models
|
|
2. In fact, Fucoidan is first protected and then treated with cisplatin, lacking evaluation of the therapeutic effect of Fucoidan. |
Thank you for the constructive advice. Initial evaluations of fucoidan will be aimed at preventing ototoxicity in our study, in most cases, prevention is better than cure. We have made the necessary changes in the revised manuscript as shown below
Pages 12, lines 362-363 The main focus of our study was to initially evaluate the protective effects of fucoidan in preventing ototoxicity rather than as a therapeutic treatment. |

Reviewer 3 Report
The topic of this manuscript is continuous research related to the antioxidative and anti-apoptotic effects of natural components in ototoxicity in mouse cochlear cells. Cisplatin-induced ototoxicity is a serious concern in clinical practice. Addressing this problem by treating it with natural material, Fucoidan was investigated by the authors' team. Some concerns are requested to be addressed before considering acceptance.
1. Fucoidan is a commercial product also made by Da Yi Biotech & Health Food. Co., Ltd. The interests between the authors' team and the biotech company need to be clarified.
2. For all western blot figures, authors should provide no spliced-together images to illustrate the results. All experimental samples and controls used for one comparative analysis should be run on the same blot/gel image, which could be provided in the supplementary material.
3. Based on the present result, the treated dose of Fucoidan was 2 mg/mL in the cell-based experiments. However, the final subjects are human beings. For humans, is there any limitation on developing Fucoidan as a health supplement to reduce Cisplatin-induced ototoxicity?
4. The findings of in vitro studies exhibited somewhat positive effects on easing ototoxicity. However, in vivo study is essential for this study. Hope the authors could present the aminal study result to support the current conclusion.
5. elaborate on reference #1.
6. Fucoidan is a kind of extract, therefore, the quantification of active components should provide in the MS.
Author Response
Response to reviewer 3:
|
Comments |
Responses |
|
1. Fucoidan is a commercial product also made by Da Yi Biotech & Health Food. Co., Ltd. The interests between the authors' team and the biotech company need to be clarified. |
Thank you for your valuable suggestions. We have made the necessary declarations in the revised manuscript.
Page 16, line 551 The authors have no conflicts of interest to disclose |
|
2. For all western blot figures, authors should provide no spliced-together images to illustrate the results. All experimental samples and controls used for one comparative analysis should be run on the same blot/gel image, which could be provided in the supplementary material. |
Thank you for the constructive advice. We have provided it in the text and supplementary material.
Page 14-15, line 459-462 Images were quantified by densitometry analysis using Image J software in version 1.54a (National Institutes of Health) and normalized to the corresponding level of β-actin in each sample. Fold change values were then calculated as the relative abundance of a target protein to the control sample on the same membrane.
Supplementary material Please see supplementary Information for details |
|
3. Based on the present result, the treated dose of Fucoidan was 2 mg/mL in the cell-based experiments. However, the final subjects are human beings. For humans, is there any limitation on developing Fucoidan as a health supplement to reduce Cisplatin-induced ototoxicity? |
Thank you for the constructive advice, we have implemented the necessary changes in the revised manuscript, as outlined below.
Page 12, line 330-357 Fucoidan, which has been used as a dietary supplement for years, has received regulatory approval from various global authorities for use in foods and supplements. Specifically, fucoidan extracts from Fucus vesiculosus have been deemed "Generally Recognized as Safe" by the US FDA and have been approved as novel foods by the European Commission. These extracts have also been approved for use in Canada and Australia in listed medicines [57]. The recommended daily consumption is up to 250 mg/day (4.17 mg/kg bodyweight (bw)/day for a 60 kg person) [79]. The study focuses on the use of fucoidan extracts from Fucus vesiculosus as a potential treatment for cisplatin-induced ototoxicity. For research about the dose of fucoidan in vitro and in vivo, in an in vivo study conducted on Sprague-Dawley rats, no significant side effects were observed when administering 0-1000 mg/kg of fucoidan orally. However, a significant elevation in plasma ALT was observed when the concentration was increased to 2000 mg/kg [80]. In the study by Naoki et al., the participants consumed 5 capsules containing 166 mg of fucoidan daily for a period of up to 12 months. No significant adverse reactions were reported among any of the participants [81]. In a study by Hsiang et al., the effectiveness of fucoidan as a complementary therapy for chemotherapy and targeted drugs in patients with metastatic colorectal cancer was evaluated. Patients consumed 4 g of fucoidan daily showed a significant improvement in disease control rate [82]. Fucoidan is a natural product that can vary in chemical composition based on the species and extraction method, so the quality and purity should be carefully evaluated before use. However, pregnant women, breastfeeding women, and those with poor liver and kidney function should not use it. A small number of people may experience mild diarrhea, but this will disappear after a few days of consumption. Fucoidan may also have blood thinning properties, so patients with poor coagulation function and anticoagulants should not use it, and it should be stopped before surgery. The iodine content of fucoidan extract is much lower than that of seaweed itself, but people who need to limit iodine intake still need to pay attention. Future studies are needed to determine the safety and efficacy of fucoidan to reduce cisplatin-induced ototoxicity.
|
|
4. The findings of in vitro studies exhibited somewhat positive effects on easing ototoxicity. However, in vivo study is essential for this study. Hope the authors could present the animal model study result to support the current conclusion. |
Thank you very much for the valuable comment. Indeed, it would be intriguing to investigate the effects of fucoidan in future studies. It is crucial to note that our results obtained in an in vitro model should be confirmed using animal or ex vivo models. Our future research will focus on validating these results using animal models. We agree with the reviewer that the lack of animal experiments is a limitation of this study, and we have elaborated on the limitation in the revised manuscript.
Pages 11, lines 253-254, page 12-13, lines 358-367 To our knowledge, this study is the first to investigate the potential otoprotective effects of fucoidan on mouse cochlear cells.
Our findings suggest that fucoidan may have a protective effect against cisplatin-induced ototoxicity, but this was only observed in the UB/OC-2 cell line. Therefore, further research is needed to confirm these results. Our future studies will aim to determine the safety and efficacy of fucoidan in reducing cisplatin-induced ototoxicity using animal models. The main focus of our study was to initially evaluate the protective effects of fucoidan in preventing ototoxicity rather than as a therapeutic treatment. In this study, we used a commercial fucoidan product that was 85% pure. We plan to explore the protective effect of fucoidan with different molecular weights, different fucoidan species, and levels of purity. Additionally, it is important to consider the most effective and safe way to administer the drug, which will be further studied using animal models |
|
5. elaborate on reference #1. |
Thank you for your valuable suggestions. We have made the necessary changes in the revised manuscript as shown below
Page 1, line 36-39 Hearing loss is a widely recognized common sensory deficit that affects a growing percentage of the world's population. According to the World Health Organization (WHO), by 2050, it is estimated that 2.5 billion people will have some degree of hearing loss, with at least 700 million requiring rehabilitation. |
|
6. Fucoidan is a kind of extract, therefore, the quantification of active components should provide in the MS. |
Thank you for your valuable suggestions. we have implemented the necessary changes in the revised manuscript.
Page 3, line 102-107 F. vesiculosus is a potent source of beneficial compounds, including fucoidans, polyphenols, fucoxanthin, and essential minerals [56, 57]. Fucoidans, a type of polysaccharide found exclusively in brown algae, are particularly notable[57, 58]. Polysaccharides from F. vesiculosus, which have a primary monomer unit of α-1,3 or α-1,4 L-fucopyranosyls, mainly consist of fucose, sulfate, and ash [59, 60] [61, 62]
Page 13, line 376-384 Commercial fucoidan extracts used in this experiment derived from Fucus vesiculosus were purchased from Da Yi Biotech & Health Food Co., Ltd. (Chiayi, Taiwan), which has good manufacturing practice certification. The raw materials have received the Australian Organic Certification (ACO: 11011) and the products also have the Taiwan Compassionate Organic Certification (TOC-AP0003 GMP) and ISO 9001 certification. Each bag contains 1 gram of powder, with a fucoidan purity of 85% and an approximate sodium content of 0.011 grams. It also contains 0.7 grams of carbohydrates, 0.05 grams iodine and 0.02 grams of protein. The main chemical components are fucose, xylose, sulfate, and ash. |

Reviewer 4 Report
The authors have checked the potential of fucoidan as an anti-oxidant and anti-apoptotic agent to mitigate ototoxicity in mouse cochlear cell lines. The authors have designed and performed the study appropriately. However, I have some major concerns regarding the study
Minor Comment
Line 68: Please remove the subscript reference 23.
Major Comments:
The authors have shown the protective effect of the fucoidan as compared to therapeutic therefore making the claims based solely on that point and also not using an animal model to test the hypothesis is selling the overall discussion made by the authors more to be desired.
The dosage used for the cell is 2mg/ml but translated to humans this could be very higher. In vivo investigation and using metabolomics and microbiome data can give more comprehensive narrative regarding the actual effect of that dosage on the physiology overall.
The study is also missing some control groups which is the major caveat in the manuscript. Using control group is a basis to compare the data in a coherent manner hence needs to be addressed.
Recommendation: Major revision and acceptance only if authors address the concerns in a comprehensive manner.
Author Response
Response to reviewer 4:
|
Comments |
Responses |
|
Minor Comment Line 68: Please remove the subscript reference 23. |
Thank you for your meticulous proofreading of the manuscript. We have removed subscript reference 23. |
|
Major Comments: The authors have shown the protective effect of the fucoidan as compared to therapeutic therefore making the claims based solely on that point and also not using an animal model to test the hypothesis is selling the overall discussion made by the authors more to be desired.
|
Thank you for the constructive advice. Initial evaluations of fucoidan will be aimed at preventing ototoxicity in our study, in most cases, prevention is better than cure. Further studies about protection effects of fucoidan in metabolomics and microbiome studies will be introduced in our team. We have made the necessary changes in the revised manuscript as shown below
Pages 13, lines 362-363 The main focus of our study was to initially evaluate the protective effects of fucoidan in preventing ototoxicity rather than as a therapeutic treatment.
Thank you very much for the valuable comment. Indeed, it would be intriguing to investigate the effects of fucoidan in future studies. It is crucial to note that our results obtained in an in vitro model should be confirmed using animal or ex vivo models. Our future research will focus on validating these results using animal models. We agree with the reviewer that the lack of animal experiments is a limitation of this study, and we have elaborated on the limitation in the revised manuscript.
Pages 11, lines 253-254, page 12-13, lines 358-367 To our knowledge, this study is the first to investigate the potential otoprotective effects of fucoidan on mouse cochlear cells.
Our findings suggest that fucoidan may have a protective effect against cisplatin-induced ototoxicity, but this was only observed in the UB/OC-2 cell line. Therefore, further research is needed to confirm these results. Our future studies will aim to determine the safety and efficacy of fucoidan in reducing cisplatin-induced ototoxicity using animal models. The main focus of our study was to initially evaluate the protective effects of fucoidan in preventing ototoxicity rather than as a therapeutic treatment. In this study, we used a commercial fucoidan product that was 85% pure. We plan to explore the protective effect of fucoidan with different molecular weights, different fucoidan species, and levels of purity. Additionally, it is important to consider the most effective and safe way to administer the drug, which will be further studied using animal models
|
|
The dosage used for the cell is 2mg/ml but translated to humans this could be very higher. In vivo investigation and using metabolomics and microbiome data can give more comprehensive narrative regarding the actual effect of that dosage on the physiology overall. |
Thank you for the constructive advice, we have implemented the necessary changes in the revised manuscript, as outlined below.
Page 12, line 330-357 Fucoidan, which has been used as a dietary supplement for years, has received regulatory approval from various global authorities for use in foods and supplements. Specifically, fucoidan extracts from Fucus vesiculosus have been deemed "Generally Recognized as Safe" by the US FDA and have been approved as novel foods by the European Commission. These extracts have also been approved for use in Canada and Australia in listed medicines [57]. The recommended daily consumption is up to 250 mg/day (4.17 mg/kg bodyweight (bw)/day for a 60 kg person) [79]. The study focuses on the use of fucoidan extracts from Fucus vesiculosus as a potential treatment for cisplatin-induced ototoxicity. For research about the dose of fucoidan in vitro and in vivo, in an in vivo study conducted on Sprague-Dawley rats, no significant side effects were observed when administering 0-1000 mg/kg of fucoidan orally. However, a significant elevation in plasma ALT was observed when the concentration was increased to 2000 mg/kg [80]. In the study by Naoki et al., the participants consumed 5 capsules containing 166 mg of fucoidan daily for a period of up to 12 months. No significant adverse reactions were reported among any of the participants [81]. In a study by Hsiang et al., the effectiveness of fucoidan as a complementary therapy for chemotherapy and targeted drugs in patients with metastatic colorectal cancer was evaluated. Patients consumed 4g of fucoidan daily showed a significant improvement in disease control rate [82]. Fucoidan is a natural product that can vary in chemical composition based on the species and extraction method, so the quality and purity should be carefully evaluated before use. However, pregnant women, breastfeeding women, and those with poor liver and kidney function should not use it. A small number of people may experience mild diarrhea, but this will disappear after a few days of consumption. Fucoidan may also have blood thinning properties, so patients with poor coagulation function and anticoagulants should not use it, and it should be stopped before surgery. The iodine content of fucoidan extract is much lower than that of seaweed itself, but people who need to limit iodine intake still need to pay attention. Future studies are needed to determine the safety and efficacy of fucoidan to reduce cisplatin-induced ototoxicity. |
|
The study is also missing some control groups which is the major caveat in the manuscript. Using control group is a basis to compare the data in a coherent manner hence needs to be addressed. |
Thank you for the comment. In this study, we established three groups of cisplatin alone, fucoidan alone, and cisplatin with fucoidan, we regard the group of cisplatin without fucoidan as the control group. Further animal studies about protection effects of fucoidan should be evidenced using transgenic or knockout mouse models.
Pages 13, lines 364-367 We plan to explore the protective effect of fucoidan with different molecular weights, different fucoidan species, and levels of purity. Additionally, it is important to consider the most effective and safe way to administer the drug, which will be further studied using animal models. |

Reviewer 5 Report
In their manuscript "Antioxidant and Anti-apoptotic Activities of Fucoidan Reduce Cisplatin-Induced Ototoxicity in Mouse Cochlear UB/OC-2 Cells" the authors describe an experimental study of the effects of fucoidan on cisplatin-induced hair cell damage and its 107 potential otoprotective activity. Their findings suggest, that fucoidan may represent a potential otoprotective agent for developing strategies for reducing cisplatin-induced ototoxicity.
The manuscript is of high quality; the authors present the results, including statistical analysis following the high standards.The experimental setup was chossen very well, also the discussion and results.
In summary, this manuscript demonstrated a very well perfomed research of high interest to the specialized readership. Medical terminolgy and methods are well choosen and clearly descrided; this work provides many useful information for further research in this field.
I recommend for publication (after minor corrections regarding spell check, grammar)
Author Response
Response to reviewer 5:
|
Comments |
Responses |
|
The manuscript is of high quality; the authors present the results, including statistical analysis following the high standards. The experimental setup was chossen very well, also the discussion and results. In summary, this manuscript demonstrated a very well perfomed research of high interest to the specialized readership. Medical terminolgy and methods are well choosen and clearly descrided; this work provides many useful information for further research in this field. I recommend for publication (after minor corrections regarding spell check, grammar) |
Thank you for your valuable and positive comments. We have made the grammar revision in the revised manuscript. Thanks
|
Round 2
Reviewer 2 Report
I have no question
Author Response
Thank you very much
Reviewer 3 Report
Looking forward to your next manuscript with the outcomes of animal studies.
Author Response
Thank you very much
Reviewer 4 Report
Authors have made a sound arguments on the comments provided and worked on the manuscript as required.
Please do a spell check before submitting the final version.
Author Response
Thank you very much, we have sent our paper to a professional institution for English revision.